# Social Participation and Functional Decline: A Comparative Study of Rural and Urban Older People, Using Japan Gerontological Evaluation Study Longitudinal Data

**DOI:** 10.3390/ijerph17020617

**Published:** 2020-01-18

**Authors:** Kazushige Ide, Taishi Tsuji, Satoru Kanamori, Seungwon Jeong, Yuiko Nagamine, Katsunori Kondo

**Affiliations:** 1Department of Community General Support, Hasegawa Hospital, Yachimata, Chiba 289-1113, Japan; 2Department of Public Health, Graduate School of Medicine, Chiba University, Chuo-ku, Chiba 260-8670, Japan; kkondo@chiba-u.jp; 3Department of Social Preventive Medical Sciences, Center for Preventive Medical Sciences, Chiba University, Chuo-ku, Chiba 260-8670, Japan; tsuji.t@chiba-u.jp (T.T.); yuiko.mail@gmail.com (Y.N.); 4School of Nursing, Tokyo Women’s Medical University, Shinjuku-ku, Tokyo 162-8666, Japan; kanamori.satoru@twmu.ac.jp; 5Department of Preventive Medicine and Public Health, Tokyo Medical University, Shinjuku-ku, Tokyo 160-8402, Japan; 6Department of Community Welfare, Faculty of Health Sciences, Niimi University, Nishigata Niimi, Okayama 718-8585, Japan; k-jeong@niimi-u.ac.jp; 7Department of Family Medicine, Graduate School of Medical and Dental Sciences, Tokyo Medical and Dental University, Bunkyo-ku, Tokyo 113-8510, Japan; 8Department of Geriatric Evaluation, Center for Gerontology and Social Science, National Center for Geriatrics and Gerontology, Obu, Aichi 474-8511, Japan

**Keywords:** social participation, work, functional decline, rural, urban

## Abstract

This study compared the relationship between social participation, including work, and incidence of functional decline in rural and urban older people in Japan, by focusing on the number and types of organizations older people participated in. The longitudinal data of the Japan Gerontological Evaluation Study (JAGES) that followed 55,243 individuals aged 65 years or older for six years were used. The Cox proportional hazards model was employed to calculate the hazard ratio (HR) of the incidence of functional decline over six years and the stratification of rural and urban settings. In this model, we adjusted 13 variables as behavioral, psychosocial, and functional confounders. The more rural and urban older people participated in various organizations, the more they were protected from functional decline. Participation in sports (HR: rural = 0.79; urban = 0.83), hobby groups (HR: rural = 0.76; urban = 0.90), and work (HR: rural = 0.83; urban = 0.80) significantly protected against the incidence of decline in both rural and urban areas. For both rural and urban older people, promoting social participation, such as sports and hobby groups and employment support, seemed to be an important aspect of public health policies that would prevent functional decline.

## 1. Introduction

A strategy of active ageing [1], by linking the key policy domains of employment, pension, retirement, health, and citizenship, provides a sound basis to respond to the challenges presented by population ageing [2]. In recent times, the practical challenge has been to open up the innovative policy spaces that might make active ageing not only thinkable but also achievable [3]. In European countries, the Active Aging Index (AAI) tool measures the untapped potential of older people for active and healthy aging across countries [4]. The AAI consists of four domains: employment, participation in society; independent, healthy, and secure living; and capacity and enabling environment for active ageing [4,5].

The social participation of older people, one of the domains of AAI [4,5], is a key factor of “successful aging” [6] and an important element of “active aging” [7]. Social participation is one of the core indicators of “age-friendly cities” proposed by the World Health Organization in recent years [8]. When considering public health services in a rapidly aging society, social participation is highlighted as a modifiable target of health interventions.

In many longitudinal studies, social participation has been reported as effective for health outcomes such as functional disability [9,10,11,12,13], cognitive disability [14,15,16], instrumental activities of daily living decline [17,18,19], and basic activities of daily living decline [20]. Among them, some studies focused on the number and types of organizations in which older people participated [10,13,16,17,18]. These studies suggested that older people who participate in more organizations are healthier than those who do not participate, and that the relationships between social participation and health varied according to the types of organization they participate in.

However, these studies [10,13,16,17,18] do not consider two issues. First, these studies do not consider “work” as a kind of social participation. According to the Organization for Economic Co-Operation and Development (OECD) scoreboard for older workers, older workers aged 65–69 years increased from 20.3% to 25.5% between 2006 and 2016 and from 12.0% to 14.6% at age 70–74 years during the same period, in OECD member countries [21]. However, recommendations made by the Council on Aging and Employment Policies [22] encourage supporting the employment of older people. In AAI, work and social participation are separate domains, but these are also defined as the actual experience of active aging [4,5]. Considering these circumstances, it is necessary to include work in social participation in the analysis of the number and types of organizations in which older people participated. Second, these studies do not consider residential environments, such as rural and urban areas. Generally, older people living in rural areas suffer with more depression [23], lower levels of basic activities of daily living [20], and a higher risk of developing disability [24] compared to those living in urban areas. In addition, the life expectancy is shorter in rural areas, and the difference between urban and rural areas is widening [25]. Previous studies [13,26] showed that rural older people were less socially active than urban older people. However, previous studies state that bonding social capital comprising a connection between community members is often stronger among rural older adults, resulting in community strength [27,28]. In addition, environmental factors such as neighborhood socioeconomic status and access to services and transportation differ in rural and urban areas [29]. Furthermore, the relationship between social participation and health outcomes, such as depression [23] and self-rated health [26,30], also differs. These may be the reasons why rural older people are unhealthier than urban older people. The relationship between social participation and functional decline may differ between rural and urban areas; however, such a relationship has not been clarified.

Thus, we aimed at clarifying whether there were differences between rural and urban areas in the relationship between social participation, including work, and incidence of functional decline. This study was conducted to inform public health policies that could prevent the need for long-term care of older individuals residing in rural and urban areas, which have different environmental factors.

## 2. Materials and Methods

### 2.1. Data

We used longitudinal data from the Japan Gerontological Evaluation Study (JAGES). JAGES is one of the few population-based gerontological repeated surveys in Japan focused on the social determinants of health and the social environment [31,32]. From August 2010 to January 2012, self-reported questionnaires were mailed to 95,827 community-dwelling independent individuals aged 65 years and older who were not eligible to receive benefits from public long-term care insurance services. They were randomly selected from 13 municipalities, including rural and urban areas. Overall, 62,418 people participated (response rate, 65.1%) in the survey called JAGES2010. They were followed up for about six years (minimum 5.2 years; maximum 6.4 years). Of the total respondents (response rate 65.1%), 54,539 (87.4%) were successfully linked to the incident records of long-term care insurance certification. We excluded 7223 responses for the following reasons: (i) missing information on address (*n* = 101) and activities of daily living (ADL) (*n* = 1482); and (ii) having physical or cognitive disabilities reported in their questionnaires (*n* = 988). Moreover, 4662 respondents in long-term care within two years were removed to avoid the possibility of reverse causality (i.e., the possibility that people who were at high risk of functional decline did not participate socially). The final number of participants in this analysis was 47,306.

Ethical approval for the study was obtained from the Nihon Fukushi University Ethics Committee (application number: 10–05), the National Center for Geriatrics and Gerontology (application number: No. 992-2), and Chiba University Ethics Committee (application number: No. 2493).

### 2.2. Dependent Variable

The dependent variable was the incidence of functional decline during the follow-up period. The incidence of functional decline was defined by medical certification for Long-Term Care Insurance. Certification of decline is based on the formal evaluation of the need for Long-Term Care according to uniform criteria applied throughout Japan, and comprises both a home-visit interview, as well as the written opinion of a primary physician [33]. This formal evaluation is based on a standardized multistep assessment of functional and cognitive impairments [33]. We obtained information on the certification of long-term needs, death, and moving out of the study area, from the long-term care insurance database maintained by the municipalities. These criteria for determining the onset of functional decline have been used in previous epidemiological studies [4,5,6].

### 2.3. Independent Variable

The independent variable was social participation. With reference to previous research [5], social participation was classified into the following six types: neighborhood groups (local community), hobby groups (hobby), sports groups or clubs (sports), industrial groups (industry), volunteer groups (volunteer), and senior citizen clubs (citizen). Furthermore, we considered work (work) as a form of social participation and therefore analyzed seven types of organizations in this study.

Participation in organizations other than work was assessed by using the following question: “How often do you participate in the following clubs or groups?”. Participants were given the following choices: “almost every day”, “twice or thrice a week”, “once a week”, “once or twice a month”, “a few times a year” and “never”. The response was categorized as “yes” if individuals selected any of the five options from “a few times a year” to “almost every day”, and “no” if they selected “never”. Participation in work was assessed by using the following question: “What is your current working status?”. Participants were given the following choices: “working”, “retired and not working now” and “never had a job”. The response was categorized as “yes” if the participants answered “working”, and “no” if they answered “retired and not working now” or “never got a job”.

The total number of types of organizations each participant participated in was tallied, and participation was categorized as 0 (no participation), 1, 2, or ≥3 organizations, or “missing”. If the response to participation in all organizations was missing, we deemed it as “missing” category in this analysis. The organizations particularly unique to Japan among the types named above are senior citizen clubs. Japan’s senior citizens clubs conduct a wide range of activities, including group activities such as sports, hobbies, cultural activities, and performing arts.

### 2.4. Covariates

Based on a previous study [10], sex, age, annual equivalized income, educational attainment, marital status, and self-reported medical conditions were considered potential confounding factors that may correlate with social participation and incidence of functional disability. In addition, behavioral, psychosocial, and physiological factors were also used as covariates and potential mechanisms influencing health and social participation. Smoking, alcohol consumption, daily walking time, and frequency of going outdoors were assessed as behavioral factors. Depression (Geriatric Depression Scale), emotional support, instrumental support, and frequency of meeting friends were assessed as psychosocial factors. Instrumental activity of daily living (IADL) was assessed as a functional factor. All variables were categorized as shown in Table 1 and set as dummy variables. A “missing” category was used in the analysis to account for missing responses to questions.

### 2.5. Classification of Rural and Urban Settings

The definition of functional urban areas from the OECD metropolitan database [34] was used to classify rural and urban settings. The definition of urban areas in OECD countries uses population density to identify urban cores, and travel-to-work flows to identify the hinterlands whose labor market is highly integrated with the cores. The methodology consisted of the three following main steps: (1) identification of core municipalities through gridded population data; (2) connecting noncontiguous cores belonging to the same functional urban area; and (3) identification of urban hinterlands. This methodology makes it possible to compare functional urban areas of similar size across different countries and classifies functional urban areas according to population size into the following four types: (1) small urban areas (with a population below 200,000 people); (2) medium-sized urban areas (with a population between 200,000 and 500,000); (3) metropolitan areas (with a population between 500,000 and 1.5 million); and (4) large metropolitan areas (with a population of 1.5 million or more). From the OECD’s four functional urban areas, all cities, including metropolitan and large metropolitan areas, were designated urban areas in this study, and all others were designated rural areas. In this study, eight municipalities were designated as rural, and five municipalities were designated as cities.

### 2.6. Statistical Analysis

First, we conducted a chi-square test to compare variables between rural and urban males and females. As the sample size in this study is very large, we calculated Cramer’s V as the effect size in addition to the *p*-value. The criteria for Cramer’s V are 0.1 for small, 0.3 for medium, and 0.5 for large. Second, the Cox proportional hazards model was employed to calculate the hazard ratios (HRs) and 95% confidence intervals (CI) of the incidence of functional decline over six years, stratified by rural, and urban settings. In each model, nonparticipation in an organization was set as the referent category. In the analysis of the number or types of organization older people participate in, we conducted a trend test. Further, six types of social participation were introduced in each model separately. The following two models of analysis were used: a regression analysis was performed with simultaneous forced entry of sex, age, equivalent income, educational attainment, marital status, and self-reported medical conditions as covariates (Model 1). Model 2 added the following confounding factors to Model 1: smoking, alcohol consumption, walking time, frequency of going outdoors, Geriatric Depression Scale, emotional support, instrumental support, frequency of meeting friends, and IADL. Finally, to confirm the robustness of our finding, we performed a complete case analysis, excluding patients missing any of the variables used in the analysis. STATA V.15 (Stata Corp, College Station, TX, USA) was used to conduct a statistical analysis, with a significance level of 5%.

## 3. Results

Table 1 presents the descriptive statistics of rural and urban variables. Of the 47,306 respondents included in the analyses, 21,921 were male and 25,385 were female. Of the 21,921 males, 6758 lived in rural and 15,163 lived in urban settings. Of the 25,385 females, 8375 lived in rural and 17,060 lived in urban settings. The average age of the rural and urban older people was 73.8 (standard deviation (SD), 5.9) and 73.3 (SD, 5.6) years, respectively. Of the respondents in rural and urban areas, 2399 (15.9%) and 4018 (15.3%) reported functional decline, respectively. The average tracking period was 2028.1 days (SD = 364.1) for rural and 1951.8 days (SD = 361.7) for urban older peoples. The comparison of variables across rural and urban areas revealed that there were many urban–rural differences. However, the sample size for this study was so large that even minor differences could result in statistical differences. In fact, Cramer’s V in the chi-square test between almost all variables was judged to be very small, and the realistic effect size was small. However, describing the difference between rural areas and cities when the effect size is 0.1 or more indicated that older people in rural areas had a lower equivalent income (*p* < 0.001), lower educational attainment (*p* < 0.001), and went outdoors less frequently (*p* < 0.001) than those in urban areas. Although the effect size was small, the distribution of the number of organizations in which older people participated differed between rural and urban areas (*p* < 0.001). When types of social participation were analyzed, rural older people participated a lot more in senior citizen clubs (*p* < 0.001) than urban older people. The distribution of participation in work differed between rural and urban areas (*p* < 0.001; Cramer’s V = 0.1); it was thought to be due to the missing category.

Table 2 presents the results of a Cox proportional hazards model analysis of the different types of organizations and incidence of functional decline. In the crude model and Model 1, a “dose–response” relationship was seen both among rural and urban areas, with progressively lower HRs as the number of different types of organizations increased. In Model 2 for rural older people, the HRs were 0.94 (95% CI: 0.84–1.05) for participation in one, 0.85 (0.75–0.97) for participation in two, and 0.76 (0.67–0.86) for participation in three or more different types of organizations, with the significant difference disappearing only for participation in one type of organization. For urban older people, the HRs were 0.92 (95% CI: 0.85–0.99) for participation in one, 0.87 (0.80–0.96) for participation in two, and 0.82 (0.75–0.89) for participation in three or more different types of organizations, with the statistical significance for one or more different types of organizations. In other words, older people in urban areas were protected from functional decline through one type of participation. On the other hand, older people in rural areas required more than one type of participation, but older people in rural areas had lower HRs when participating in more than two types of organizations than older people in urban areas.

The results of the complete case analysis are shown in Appendix A. The results of the complete case analysis, excluding patients missing any of the variables used in the analysis were similar to those when the “missing” category was used in the analysis to account for missing responses to questions. The full modeling results in Model 2 were presented in Appendix A. In this study, Model 2 added social networks such as emotional support, instrumental support, and frequency of meeting friends. Rural and urban older people who could not avail emotional support were not protective against functional decline compared with those who could avail it. Furthermore, urban older people who could not avail instrumental support were not protective against functional decline compared with those who could avail it, but this was not the case in rural areas. The frequency of meeting friends was not statistically significant in rural and urban areas.

Table 3 presents the results of the Cox proportional hazards model analysis of the type of social participation and incidence of functional decline. Almost all types of organizational participation were strongly protective against functional decline, but senior citizen clubs had the opposite relationship in the crude model. Similarly, many types of organizational participation were protective against functional decline in Model 1. In Model 2 for rural older people, participation in hobbies (HR = 0.76; 95% CI: 0.68–0.85), sports (HR = 0.79; 95% CI: 0.69–0.89), work (HR = 0.83; 95% CI: 0.76–0.91), and local community (HR = 0.86; 95% CI: 0.77–0.95) was found to be protective against the incidence of decline. For urban older people, participation in work (HR = 0.80; 95% CI: 0.70–0.91), sports (HR = 0.83; 95% CI: 0.77–0.91), and hobbies (HR = 0.90; 95% CI: 0.84–0.97) was found to be protective against the incidence of decline.

The results of the complete case analysis are shown in Appendix A. In the complete case analysis for rural older people, the HR for participation in work and local community was below 1.00, but the statistical significance disappeared. Further, the results of the complete case analysis for urban older people, excluding patients missing any of the variables used in the analysis, were similar to those when the “missing” category was used in the analysis, to account for missing responses to questions. The full modeling results in Model 2 were presented in Appendix A. The result of the social networks, such as emotional support, instrumental support, and frequency of meeting friends, added in Model 2, was similar to the analysis of the number of organizations.

## 4. Discussion

To the best of our knowledge, this is the first longitudinal study to compare the relationship between the number and type of organizations, including work, and incidence of functional decline in rural and urban areas separately.

In all, two findings were obtained from this study: (1) a “dose–response” relationship was seen both among rural and urban areas, with progressively lower HRs as the number of different types of organizations increased; and (2) participation in sports, hobbies, and work were protective against incidences of decline in both rural and urban areas. In this study the classification of rural and urban areas is as proposed by the OECD. Previous studies have used population density and national classification; however this study adopted an international classification system. Even when classifying areas by population density, the results of this study were almost the same.

The analysis of the number of organizations revealed the HRs of the number of types of organizations progressively decreased as the number of participating organizations. This supports previous studies, including those measuring other health outcomes [10,13,16,17,18]. In this study, HRs were lower when participated in two or more types of organizations in rural areas than in urban areas. The social relationships specific to rural areas may be the reason why older people living in these areas need to join more than one organization. Previous studies have indicated that bonding social capital comprising a connection between community members is often stronger among rural older adults, resulting in community strength [27,28]. In this study, participation in local community organizations was higher in rural areas than in urban areas. However, excessive bonding social capital tends to have negative effects [35]. According to the systematic review of the negative health effects of social capital [35], there are downsides to social capital that emerge in the context of strong bonding social capital, but not in weak bridging social capital. (1) Strong bonding ties impose heavy obligations on community members by following a dominant social hierarchy and social norms, and it exclude outsiders. (2) The lack of bridging SC is crucial in socioeconomically disadvantaged communities. (3) In such settings, the connection of members to outside sources of support is even more important. In closed communities, such as those in rural areas, participation in more organizations may improve bridging social capital. For the above reasons, older people in rural areas may benefit from participation in a greater number of organizations.

Many types of organizational participation were protective against functional decline in Model 1, which included factors such as age, equivalent income, educational attainment, marital status, and self-reported medical conditions. However, in Model 2, factors such as behavioral, psychosocial, and functional confounders, as well as participation in sports, hobbies, and work, were protective against incidences of decline.

This is the first longitudinal study to compare work with other community organizations by defining work as a type of social participation. Previous longitudinal studies focusing on the relationship between work and health outcomes among older people have examined work alone [20,36,37], and comparisons with other community organizations have been cross-sectional studies [38]. Given the current challenges posed by a rapidly declining birthrate and aging population, it is necessary to develop a social structure where many older people work [39]. Working support for older people is expected to contribute not only to a substantial increase in the labor force but also to a decrease in the number of older people requiring care [39]. This study showed that working support and improvement of working environment could be public health policies that would prevent the need for the provision of long-term care in rural and urban areas. In longitudinal studies of work and health outcomes [20,36,37], work is generally considered good for health. However, poor-quality work [36] is not good for health; therefore, additional analysis is necessary, since this research did not consider the type of work.

The results indicating that participation in sports and hobby groups were protective against disability were similar to previous studies [10]. According to this result, a good public health policy would include local government provision of regular opportunities for social participation in sports and hobbies in both rural and urban areas. Participation in sports and hobby groups has also been reported to prevent other poor health outcomes [16,18,19]. Previous studies have shown that older people in rural areas were unhealthier than older people in urban areas [20,24,29,30], but in this study, there was no difference in the incidence of functional disability between older people in rural and urban areas. The reason may be that there was no difference in the urban–rural participation rates in sports and hobby groups among the participants of this study. Even in rural areas, promoting participation in groups such as sports and hobbies groups could prevent the incidence of functional decline. In Japan, the salon-type community intervention [40] has been implemented as one of the ways to promote social participation. These salons, managed by local volunteers, are held once or twice a month in communal spaces within walking distance of community members’ homes, and older people can meet and interact with others through enjoyable, relaxing, and sometimes educational programs [40]. Moreover, participation in local community organizations was only protective against decline in rural older people in this study. In the scoping review by Carver et al. [41], older people who lived in rural areas had many opportunities to engage in community-association activities, and through such social participation, a sense of belonging was created. They suggested that such social participation is important for achieving successful aging in rural areas.

The preventive effects against functional decline of the number and types of social participation were almost the same among both rural and urban areas in this study. Quite a few studies on age-friendly cities showed that urban areas are suitable for active aging [42]. One of the reasons for that may be a larger quantity of amenities and possibilities for social interactions/organizations and easier access to those in urban areas compared to rural areas. It could be an effective intervention that older people move to urban areas when their health/physical ability starts to decline. Nowadays, the compact city, i.e., located in the rural city center, but short-distance from urban functions, trials have begun in Japan [43,44], and their effectiveness is expected to be verified.

This study has two strengths. First, this is the first study to target older people in many municipalities, including rural and urban areas, in contrast to previous studies that only focused on the number and type of organization in which older people participated [10,13,16,17,18]. Second, the data used in this study was collected over a long period (about six years) and excluded respondents receiving long-term care within two years, thereby removing the possibility of reverse causality (i.e., the possibility that people who had a high risk of functional decline did not participate socially).

This study had three limitations. First, we did not consider the frequency of social participation. It has been reported that the relationship between social participation and health outcomes differs depending on the frequency of social participation [16,18,38]. However, this research emphasized a comparison between rural and urban areas in line with a previous study [10]. Second, we did not consider the older people’s role of the organization they participated in, such as being a member or a leader. A leading role in an organization has an additional effect on social participation and health outcomes [15,45]. Finally, this study only focuses on the differences between rural and urban areas; however, there may be other environmental characteristics to consider. The NuAge Study showed environmental factors associated with social participation of older people vary by living areas, such as metropolitan, urban, and rural areas [29]. In Japan, it was reported that environmental factors such as access to facilities, shops, and parks and sidewalks were related to participation in sports groups [46]. Future longitudinal or interventional studies focusing on rural and urban environmental improvement will be needed.

## 5. Conclusions

We compared the relationship between social participation, including work and incidence of functional decline in rural and urban older people, to inform public health policies that would prevent functional decline in older individuals residing in Japan. Participating in various organizations protected older people from functional decline, and, thus, it might be essential to facilitate the benefits of such participation to both rural and urban older people. Furthermore, participation in sports, hobbies, and work was protective against incidences of decline in both rural and urban regions. For both rural and urban older people, promoting social participation, such as sports and hobbies groups and employment support, seems to be an important aspect of public health policies that would prevent functional decline.

## Figures and Tables

**Table 1 ijerph-17-00617-t001:** Baseline characteristics of respondents (2010–2012).

Variables		Rural (*n* = 15,083)		Urban (*n* = 32,223)		*p*-Value (Cramer’s V)
		*n*	%	*n*	%	
Sex	Male	6758	44.8	15,163	47.0	<0.001 (0.02)
	Female	8325	55.2	17,063	53.0	
Age (years)	65–69	4287	28.4	9676	30.0	<0.001 (0.04)
	70–74	4536	30.1	10,343	32.1	
	75–79	3470	23.0	7257	22.5	
	80–84	1973	13.1	3611	11.2	
	85+	817	5.4	1336	4.2	
Equivalent income (million yen)	Low (<2.0)	7127	47.3	11,815	36.7	<0.001 (0.13)
	Middle (2.0–3.9)	3988	26.4	11,321	35.1	
	High (≥4.0)	939	6.2	3616	11.2	
	Missing	3029	20.1	5471	17.0	
Educational attainment (years)	<10	8404	55.7	14,103	43.8	<0.001 (0.12)
	10–12	4432	29.4	11,182	34.7	
	≥13	1853	12.3	6255	19.4	
	Missing	394	2.6	683	2.1	
Marital status	Married	10,611	70.3	23,315	72.4	<0.001 0.04
	Single	4111	27.3	8480	26.3	
	Missing	361	2.4	428	1.3	
Self-reported medical conditions	Illness	10,267	68.1	21,859	67.8	0.243 (0.01)
	No illness	3508	23.2	7677	23.8	
	Missing	1308	8.7	2687	8.4	
Smoking	Never smoked	8440	56.0	17,174	53.3	<0.001 (0.04)
	Past smoker	3566	23.6	8649	26.9	
	Current smoker	1437	9.5	3330	10.3	
	Missing	1640	10.9	3070	9.5	
Alcohol consumption	Never drank	9122	60.5	18,420	57.2	<0.001 (0.03)
	Past drinker	440	2.9	1021	3.2	
	Current drinker	4684	31.1	10,836	33.6	
	Missing	837	5.5	1946	6.0	
Walking time (per day)	>90 min	2515	16.7	5004	15.5	<0.001 (0.05)
	60–90 min	2130	14.1	5029	15.6	
	30–60 min	4558	30.2	10,960	34.0	
	<30 min	4905	32.5	9341	29.0	
	Missing	975	6.5	1889	5.9	
Frequency of going outdoors	Almost everyday	6607	43.8	18,496	57.4	<0.001 (0.15)
	2–3 times/week	4601	30.5	8288	25.7	
	About once/week	1755	11.6	2253	7.0	
	Rarely	1351	9.0	1384	4.3	
	Missing	769	5.1	1802	5.6	
Depression	No depression	9313	61.8	19,977	62.0	0.002 (0.02)
	Depressive tendency	2613	17.3	5343	16.6	
	Depression	822	5.4	1595	4.9	
	Missing	2335	15.5	5308	16.5	
Emotional support	Available	13,052	86.5	28,584	88.7	<0.001 (0.03)
	Not available	1209	8.0	2310	7.2	
	Missing	822	5.5	1329	4.1	
Instrumental support	Available	12,369	82.0	26,971	83.7	<0.001 (0.03)
	Not available	1906	12.6	3906	12.1	
	Missing	808	5.4	1346	4.2	
Frequency of meeting friends	Almost everyday	2247	14.90	4347	13.5	<0.001 (0.04)
	2–3 times/week	3513	23.30	7206	22.4	
	About once/week	2461	16.30	5293	16.4	
	1–2 times/month	2914	19.30	6049	18.8	
	A few times a year or less	2841	18.80	7251	22.5	
	Missing	1107	7.40	2077	6.4	
Instrumental activity of daily living (IADL)	Not decline	6073	40.3	12,578	39.0	0.025 (0.01)
	Decline	7230	47.9	15,856	49.2	
	Missing	1780	11.8	3789	11.8	
Number of types of organizations	0	3187	21.1	7700	23.9	<0.001 (0.06)
	1	3420	22.7	7964	24.7	
	2	2691	17.8	6173	19.1	
	≥3	5227	34.7	9588	29.8	
	Missing	558	3.7	798	2.5	
Type of social participation						
Local community	Nonparticipation	5909	39.2	14,781	45.9	<0.001 (0.07)
	Participation	5886	39.0	10,802	33.5	
	Missing	3288	21.8	6640	20.6	
Hobby	Nonparticipation	6656	44.1	13,411	41.6	<0.001 (0.05)
	Participation	5430	36.0	13,212	41.0	
	Missing	2997	19.9	5600	17.4	
Sports	Nonparticipation	8540	56.6	18,242	56.6	0.001 (0.02)
	Participation	3272	21.7	7375	22.9	
	Missing	3271	21.7	6606	20.5	
Industry	Nonparticipation	8791	58.3	19,293	59.9	<0.001 (0.04)
	Participation	2020	13.4	4829	15.0	
	Missing	4272	28.3	8101	25.1	
Volunteer	Nonparticipation	8787	58.3	19,803	61.5	<0.001 (0.03)
	Participation	2292	15.2	4526	14.0	
	Missing	4004	26.5	7894	24.5	
Citizen	Nonparticipation	7842	52.0	19,244	59.7	<0.001 (0.10)
	Participation	4247	28.2	6165	19.1	
	Missing	2994	19.8	6814	21.2	
Work	Nonparticipation	9002	59.7	21,572	67.0	<0.001 (0.10)
	Participation	3397	22.5	7166	22.2	
	Missing	2684	17.8	3485	10.8	

**Table 2 ijerph-17-00617-t002:** HRs for participation in one, two, and three or more different types of organizations.

**Rural**	**Crude Model**	**Model 1**	**Model 2**
	**HR (95% CI)**	**HR (95% CI)**	**HR (95% CI)**
0	1.00 Ref	1.00 Ref	1.00 Ref
1	0.71 * (0.64–0.79)	0.87 * (0.78–0.97)	0.94 (0.84–1.05)
2	0.57 * (0.50–0.64)	0.75 * (0.66–0.85)	0.85 * (0.75–0.97)
≥3	0.43 * (0.39–0.49)	0.62 * (0.56–0.70)	0.76 * (0.67–0.86)
Trend *p*	*p* < 0.05	*p* < 0.05	*p* < 0.05
**Urban**	**Crude model**	**Model 1**	**Model 2**
	**HR (95% CI)**	**HR (95% CI)**	**HR (95% CI)**
0	1.00 Ref	1.00 Ref	1.00 Ref
1	0.69 * (0.64–0.75)	0.85 * (0.79–0.91)	0.92 * (0.85–0.99)
2	0.59 * (0.55–0.65)	0.77 * (0.71–0.84)	0.87 * (0.80–0.96)
≥3	0.48 * (0.44–0.52)	0.67 * (0.62–0.72)	0.82 * (0.75–0.89)
Trend *p*	*p* < 0.05	*p* < 0.05	*p* < 0.05

HR: Hazard ratio; CI: confidence interval; Ref: reference. * *p* < 0.05. Model 1: Crude model + sex, age, equivalent income, educational attainment, marital status, and self-reported medical conditions. Model 2: Model 1 + smoking, alcohol consumption, walking time (per day), frequency of going outdoors, depression, emotional support, instrumental support, frequency of meeting friends, and IADL.

**Table 3 ijerph-17-00617-t003:** HRs for type of social participation (reference: nonparticipation in each organization).

**Rural**	**Crude Model**	**Model 1**	**Model 2**
	**HR (95% CI)**	**HR (95% CI)**	**HR (95% CI)**
Local Community	0.59 * (0.54–0.65)	0.77 * (0.70–0.86)	0.86 * (0.77–0.95)
Hobby	0.57 * (0.52–0.64)	0.61 * (0.60–0.75)	0.76 * (0.68–0.85)
Sports	0.62 * (0.55–0.70)	0.70 * (0.62–0.78)	0.79 * (0.69–0.89)
Industry	0.67 * (0.58–0.78)	0.92 (0.84–1.01)	1.01 (0.87–1.18)
Volunteer	0.62 * (0.54–0.72)	0.77 * (0.67–0.89)	0.89 (0.77–1.03)
Citizen	1.40 * (1.27–1.54)	0.94 (0.85–1.03)	1.02 (0.93–1.13)
Work	0.48 * (0.42–0.54)	0.74 * (0.65–0.84)	0.83 * (0.76–0.91)
**Urban**	**Crude Model**	**Model 1**	**Model 2**
	**HR (95% CI)**	**HR (95% CI)**	**HR (95% CI)**
Local Community	0.70 * (0.65–0.75)	0.84 * (0.79–0.90)	0.95 (0.88–1.01)
Hobby	0.70 * (0.66–0.75)	0.78 * (0.73–0.84)	0.90 * (0.84–0.97)
Sports	0.60 * (0.55–0.65)	0.73 * (0.67–0.79)	0.83 *(0.77–0.91)
Industry	0.81 * (0.74–0.89)	0.90 (0.78–1.05)	1.04 (0.95–1.15)
Volunteer	0.66 * (0.60–0.73)	0.80 * (0.73–0.89)	0.94 (0.85–1.04)
Citizen	1.37 * (1.28–1.47)	0.89 * (0.83–0.96)	0.99 (0.92–1.07)
Work	0.50 * (0.46–0.55)	0.80 *(0.73–0.87)	0.80 * (0.70–0.91)

HR: Hazard ratio; CI: confidence interval; Ref: reference. * *p* < 0.05. Model 1: Crude model + sex, age, equivalent income, educational attainment, marital status, and self-reported medical conditions. Model 2: Model 1 + smoking, alcohol consumption, walking time (per day), frequency of going outdoors, depression, emotional support, instrumental support, frequency of meeting friends, and IADL.

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
