# Peer review of "Social Participation and Functional Decline: A Comparative Study of Rural and Urban Older People, Using Japan Gerontological Evaluation Study Longitudinal Data"

_ijerph, 2020, doi:10.3390/ijerph17020617_

Round 1
Reviewer 1 Report
In this study, the authors compared the relationship between social participation, including work, and incidence of functional decline in rural and urban older people in Japan. The authors analyzed two research groups - elderly people living in rural and urban areas. The differences between participation in social life (social participation) were examined, as the also included professional work and the perception of the decline in the functionality of the elderly.
Comments:
Line number 44: Please provide more extensive literature on active aging. This basic older literature includes the works of:
Kalache A., Kickbush I., (1997), A global strategy for healthy ageing, World Health (4).Walker, A. C., (2002), A strategy for active ageing, International Social Security Review, 55 (1), 121-38.ActivAge Consortium, (2008),Overcoming the barriers and seizing the opportunities for active ageing policies in Europe, International Social Science Journal, 58 (190 December 2006), 617-31.
New UNECE literature / European Commission (2019) “2018 Active Ageing Index: Analytical Report”, Report prepared by Giovanni Lamura and Andrea Principi under contract with the United Nations Economic Commission for Europe (Geneva), co-funded by the European Commission’s Directorate General for Employment, Social Affairs and Inclusion (Brussels)
Please mention the established synthetic Active Aging Index described in the work (cf. Zaidi, A., et al., 2013) and calculated for European countries.
Zaidi, A., K. Gasior, M.M. Hofmarcher, O. Lelkes, B. Marin, R. Rodrigues, A. Schmidt, P. Vanhuysse and E. Zolyomi (2013), ‘Active Ageing Index 2012: Concept, Methodology and Final Results ‘, Methodology Report Submitted to European Commission’s DG Employment, Social Affairs and Inclusion, and to Population Unit, UNECE, for the project: ‘Active Ageing Index (AAI)’, UNECE Grant No: ECE/GC/2012/003, Geneva.
Please indicate whether such an Active Aging Index in four areas - Employment, Social participation, Independent, healthy and secure living, Capacity and enabling environment for active aging is calculated in Japan, or can it be compared with European countries.
Line number 55 First, these studies do not consider "work" as a kind of social participation.
Please specify which studies were conducted?
As in Europe, probably research in Europe includes "work -employment" as a separate area in the Active Ageing Index. In the Active Aging Index - "work - employment" is equal to 35%, which is the most important factor and is not part of the area of social activity.
I propose to improve the admission taking into account all the information given above and indicating why it was decided to analyze such variables in order to diagnose the differences between social participation which also includes professional work and the perception of the decline in the functionality of the elderly.
Line number 83 From August 2010 to January 2012, self-reported questionnaires were mailed to 95,827 community-dwelling independent individuals aged 65 years and older who were not eligible to receive benefits from public long-term care insurance services.
Please specify if this means how long the survey took from which year to which year. Is it only in 2010-2012? If so, are there more recent, other data in this area.
Line number 90 Please enter full name - ADL
Line number 138 Please enter full name – IADL
Line number 193 Table 1. Baseline characteristics of respondents
Please indicate in which years the respondents responded.
The discussion presented is very interesting. However, I suggest changing and shortening the title of the article.
Author Response
Dear Reveiwer 1
Thank you for your though consideration of our manuscript and for providing such constructive feedback.
Please see the attachiment.
Sincerely,
Kazushige Ide

Reviewer 2 Report
This manuscript studies the relationship between social participation and functional decline among older adults in Japan. The topic is timely and relevant.
Introduction: The literature review in the introduction is rather short and mainly discusses Japanese/Asian studies. The paper would benefit from a more extensive literature study with references to European and American studies.
Lines 66-69 state that rural older adults are less socially active than urban older adults. However, there are also studies that indicate that stronger local social networks exist in rural areas.
Data: A large national dataset is used which is nice. However, the quality of the data is not great as it contains a lot of missings. Instead of creating "missing" categories, I would suggest to remove cases with missing data. Still sufficient cases will remain.
Social participation is defined as participation in organizations. However, not all social participation is organized in clubs or associations. According to Table 1 the data also contain information on frequency of meeting with friends (as well as emotional and instrumental support, indicating presence of a social network). Club membership may not be necessary if people have sufficient contact with friends. I suggest to include social interaction with friends in the model as part of social participation.
The inclusion of work as social participation is fine, but not a unique strength of the paper.
The results should be explained more and the full modeling results should be shown in Table 2 and 3.
Regarding the environment, the analysis only focuses on differences between urban and rural. It would have been interesting to include other environmental characteristics, such as presence of amenities/meeting places. Lines 262-263 state that environmental improvement could be a good policy, however, it is not clear how the environment could be improved.
It is possible that older adults move to (urban) areas with more amenities (and possibilities for social interactions/organizations) when their health/physical ability starts to decline. This could be discussed.
It would also be good to discuss how (membership of) organizations could be promoted.
Author Response
Dear Reveiwer 2
Thank you for your through consideration of our manuscript and for provideing such constructive feed back.
Please see the attachment.
Sincerely,
Kazushige Ide

Reviewer 3 Report
Thank you for allowing me to review this manuscript entitled: « Does social participation have preventive relationship with functional decline?: A comparative study of rural and urban older people using Japan Gerontological Evaluation Study Longitudinal data ». The objective of the paper was to study the association between social participation and functional decline in different kinds of older populations. This topic is of high interest as avoiding functional decline is a key parameter to preserve older adults' autonomy and social participation is a crucial known factor. The manuscript is interesting and clearly written from the introduction to the conclusion. I detected no significant issue regarding the methods used and the interpretation of the results.
Author Response
Dear Reviewer 3
Thank you for your through consideration of our manuscript and for providing such constructive feed back.
Please see the attachment.
Sincerely,
Kazushige Ide

Round 2
Reviewer 2 Report
Personally, I would only include the models without missing values and i would put the full modeling results in the main body instead of in an appendix.
Other than that, my comments have been sufficiently addressed